# Inter-Golgi transport mediated by COPI-containing vesicles carrying small cargoes

**Patrina A Pellett[1,2], Felix Dietrich[1,3], Jörg Bewersdorf[1], James E Rothman[1]\*, Grégory Lavieu[1]\***

[1]Department of Cell Biology, Yale University School of Medicine, New Haven, United States; [2]Department of Chemistry, Yale University, New Haven, United States; [3]Department of Mathematics, Technische Universität München, Garching, Germany

**Abstract** A core prediction of the vesicular transport model is that COPI vesicles are responsible for trafficking anterograde cargoes forward. In this study, we test this prediction by examining the properties and requirements of inter-Golgi transport within fused cells, which requires mobile carriers in order for exchange of constituents to occur. We report that both small soluble and membrane-bound secretory cargo and exogenous Golgi resident glycosyl-transferases are exchanged between separated Golgi. Large soluble aggregates, which traverse individual stacks, do not transfer between Golgi, implying that small cargoes (which can fit in a typical transport vesicle) are transported by a different mechanism. Super-resolution microscopy reveals that the carriers of both anterograde and retrograde cargoes are the size of COPI vesicles, contain coatomer, and functionally require ARF1 and coatomer for transport. The data suggest that COPI vesicles traffic both small secretory cargo and steady-state Golgi resident enzymes among stacked cisternae that are stationary.

**\*For correspondence:** james. rothman@yale.edu (JER); gregory.lavieu@yale.edu (GL)

**Competing interests:** The authors declare that no competing interests exist.

**Reviewing editor**: Vivek Malhotra, Center for Genomic Regulation, Spain

## Introduction

The Golgi apparatus is a central feature of the secretory pathway in all eukaryotic cells. In higher eukaryotes, the Golgi stack consists of four to six flattened cisternae, which contain a series of glycosyl-transferases and other resident membrane proteins. These are localized in the order of their function in distinct steady-state distributions along the axis between the cis (entry) and the trans (exit) face, as a result of a dynamic equilibrium resulting from a balance of anterograde (ER → cis → trans) and retrograde (trans → cis and Golgi → ER) flows. Proteins secreted from the cell, as well as constituents of the plasma membrane and a broad variety of membrane-enclosed compartments pass through the ER–Golgi system in the anterograde direction, diverging only as they depart the Golgi stack at its trans face (also termed the TGN). During this anterograde passage they are typically glycosylated in a step-wise fashion as they encounter the responsible enzymes (*Emr et al., 2009*; *Klumperman, 2011*).

There are two broadly opposing alternative mechanisms (with many variations) to explain anterograde transport (cis → trans) across the Golgi (*Emr et al., 2009*; *Rothman, 2010*): (1) mobile cisternae (also termed cisternal progression), in which the cisterna themselves move from cis → trans, being continuously remodeled by retrograde flow of resident enzymes in the process (*Mironov et al., 2001*; *Losev et al., 2006*; *Matsuura-Tokita et al., 2006*). (2) The vesicular transport model, in which cisterna are viewed as static and the anterograde cargo must then be mobile, moving forward from cisterna-to-cisterna by a carrier mechanism. COPI-coated vesicles are the principal candidates for anterograde carriers, as they form and fuse copiously throughout the Golgi stack, and many contain anterograde cargo (*Balch et al., 1984b*; *Rothman and Wieland, 1996*; *Orci et al., 1997*; *Rothman, 2010*). COPI vesicles are known to carry retrograde cargo from Golgi to ER (*Letourneur et al., 1994*; *Emr et al., 2009*). However, dynamic tubular connections between cisternae have also been proposed (*Trucco et al., 2004*).

**eLife digest** All eukaryotic cells contain an organelle called the Golgi apparatus, which consists of a series of four to six flattened structures called cisternae. Proteins that are intended for secretion from the cell, or proteins that go on to become part of the cell membrane, must pass through the Golgi, where they undergo modifications that ensure they are targeted to the correct place.

There are two main models for how proteins are transported from the entry side of the Golgi, known as the cis face, to the exit side (trans face), through a process known as anterograde transport. One possibility is that the cargo protein matures within a single cisterna, which gradually moves from the cis to the trans face without the protein ever leaving it. Alternatively, the cisternae may remain fixed in position, while individual proteins are carried between them by specialized transport vesicles called COPI vesicles.

Now, Pellett et al. have used modern molecular biology techniques to revisit this question, more than 25 years after members of the same group first obtained evidence suggesting the involvement of COPI vesicles. To do this, they labelled the proteins that reside within the Golgi of one cell green, and those within the Golgi of another cell, red. They then fused the two cells together, and traced the movement of labelled proteins between the two organelles.

Proteins that are known to undergo anterograde transport were also transported between the two Golgi, whereas large protein aggregates were not. Super-resolution microscopy revealed that the transported proteins were carried in vesicles the size of COPI vesicles and surrounded by a coat protein that resembles COPI. Moreover, transport involved the adaptor protein ARF, which helps to load cargo into COPI vesicles.

By providing evidence that Golgi resident proteins and proteins that normally undergo anterograde transport can be carried by COPI vesicles between two physically separate Golgi, Pellett et al. increase the weight of evidence that COPI vesicles may also be responsible for both retrograde and anterograde transport within the Golgi itself.

In this study, we seek to gain insight into how small anterograde (and retrograde) directed cargoes traverse stacked Golgi. We also test core predictions of the COPI vesicular model for anterograde transport among static cisternae. For these purposes we have employed a simple assay for inter-Golgi transport within fused cells, which requires mobile carriers in order for exchange of Golgi constituents to occur.

Over 25 years ago, long before dynamic imaging in live cells with GFP tags was possible, we reported the surprising finding that VSV-encoded G protein is capable of rapid exchange between two different Golgi populations within fused cells (*Rothman et al., 1984a*, *b*). These results recapitulated a previous discovery made in cell-free extracts (*Fries and Rothman, 1980*; *Balch et al., 1984a*), which later allowed the identification of key vesicle transport machinery (such as coatomer, NSF, and SNAREs) needed for inter-Golgi transport (*Malhotra et al., 1988*; *Waters et al., 1991*; *Serafini et al., 1991a*, *b*).

Although the data generated with this biochemical approach strongly indicated that VSV-G containing COPI vesicles were employed in anterograde transport in the Golgi (*Orci et al., 1989*), this interpretation was debated (*Mellman and Simons, 1992*; *Pelham, 1994*) because the role of COPI vesicles in retrograde transport was subsequently uncovered (*Letourneur et al., 1994*), and the then available data on inter-Golgi traffic could not rule out that Golgi glycosyl-transferases are also transported between Golgi and could contribute an unknown portion of the assay signals (*Love et al., 1998*; *Walter et al., 1998*). Other groups used a cell fusion assay to observe the behavior of fluorescently labeled Golgi markers, but were only able to do so hours after fusion had taken place, when the Golgi were already reassembled at the center of the newly-formed poly-karyons (*Ho et al., 1990*; *Deng et al., 1992*).

Here, we have reinvestigated inter-Golgi transport with much more precise and penetrating tools of modern cell and molecular biology and optics, and find that both small anterograde cargo and Golgi resident glycosyl-transferases are briskly exchanged. Large soluble aggregates, which traverse the stack, do not transfer between Golgi, implying that small cargo (which can fit in a typical transport vesicle) are transported by a different mechanism. Super-resolution microscopy and live cell imaging

reveal that the carriers of both anterograde and retrograde cargo are the size of COPI vesicles, contain coatomer, and are Arf1-dependent.

## Results

### Exogenous Golgi resident membrane proteins traffic between Golgi in fused cells

To assess inter-Golgi exchange in real time we fused two populations of HeLa cells, one containing GFP tagged galactosyltransferase (GT-GFP) with a second containing RFP tagged sialyltransferase (ST-RFP) (*Schaub et al., 2006*) (*Figure 1*). The cells were separately transfected, combined and allowed to spread as a mixed population on a cover slip. At the steady state in HeLa cells, GT and ST have partially overlapping distributions in Golgi cisternae (*Rabouille et al., 1995*; *Schaub et al., 2006*). GT is localized in the trans Golgi cisternae, while ST is localized in the trans and trans-most/TGN cisternae. The cell population containing GT-GFP was also transfected with VSV-G protein to enable us to trigger cell–cell fusion with a brief acidic exposure (*Florkiewicz and Rose, 1984*). Following pH 5 exposure, the fused cells were immediately monitored by confocal video-microscopy, enabling us to record the time dependence of exchange of these markers between the Golgi populations in the fused cells from the onset of fusion. Importantly, all the experiments were performed at 20°C and in the presence of cycloheximide (unless mentioned otherwise), to block new synthesis of GT-GFP and ST-RFP in the ER during the experiment and to block exit of these markers from the Golgi while enabling transport in the Golgi to proceed (*Matlin and Simons, 1983*). Each Golgi area remains tightly associated with its original nucleus for at least 2 hr, longer than the maximum time course of the exchange experiments.

We observed that the Golgi originally containing the GT-GFP marker progressively acquired the ST-RFP marker (*Figure 2*, Golgi 3 and 4), the average fluorescence intensity of such Golgi areas increasing by a factor 10.2 ± 3.7 (n = 5) at 30 min post-fusion at 20°C. Reciprocally, the initially red-labeled Golgi areas acquired the GT-GFP marker (*Figure 2*, Golgi 1 and 2) with a 12.1 ± 6.1 (n = 5) fold increase at 30 min post-fusion. An interesting case arises in a polykaryon when a GT-GFP transfected cell fuses with an ST-RFP transfected cell and a third cell that failed to be transfected with either marker (*Figure 2*, Golgi 5). Here, the initially non-fluorescent Golgi simultaneously acquires markers from both red and green Golgi. Altogether, these initial studies establish that some exogenous resident Golgi enzymes traffic between separate Golgi populations in fused cells.

Most proteins require about 10–20 min to fully traverse the Golgi stack at 37°C, and this process is expected to be several fold slower at 20°C. Therefore, the time course of exchange between Golgi, remarkably enough, seems to be similar to that for transit of a Golgi stack in a single cell. Moderate

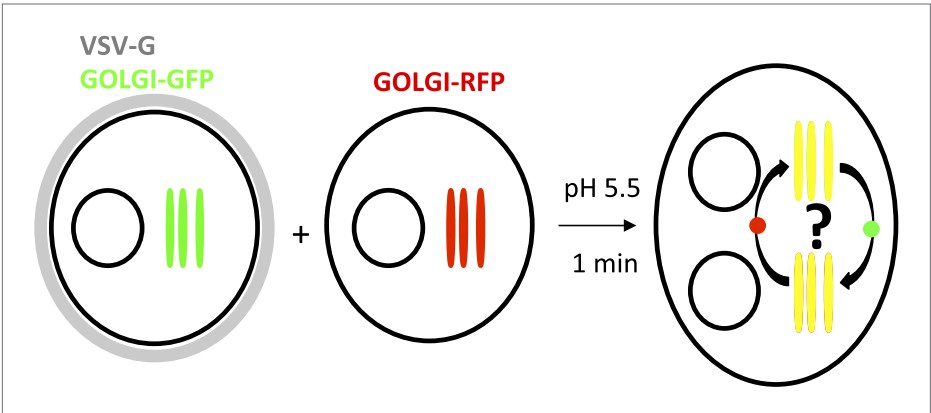

**Figure 1**. General procedure. HeLa cells co-expressing a GFP-labeled Golgi localized protein and VSV-G are mixed with HeLa cells expressing a RFP-labeled Golgi localized protein. Cell fusion is triggered by acidic exposure of cell surface targeted VSV-G. Cycloheximide was added 1 hr prior to fusing the cells and during the imaging procedure, to prevent de novo protein synthesis. Golgi-content mixing is assessed by live imaging confocal imaging, visualization and characterization of the putative transport intermediates are assessed by STED microscopy.

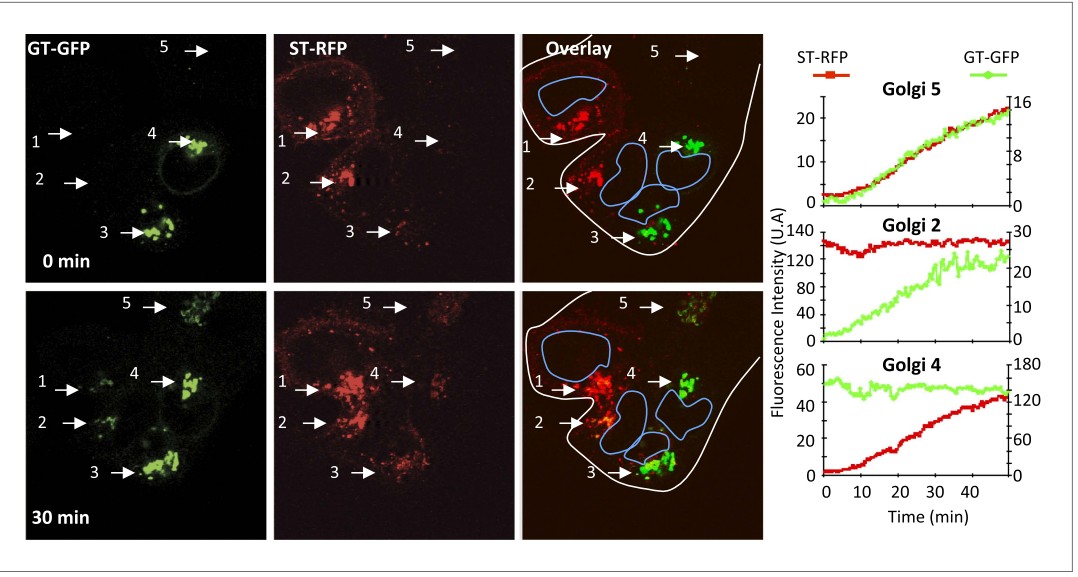

**Figure 2**. Inter-Golgi transport of Golgi resident glycosyl-transferase proteins. HeLa cells expressing either GT-GFP and VSV-G, or ST-RFP were mixed and fused by acidic exposure (1 min, pH 5) and then monitored by confocal video-microscopy at 20°C. Cells were treated with CHX (100 μg/ml) 2 hr prior to fusion and during the imaging. Graphs show fluorescence intensity of markers within Golgi 2, 4, and 5 over time. Results are representative of three independent experiments.

rate differences and variations in this exchange can be expected due to the variability of Golgi numbers per polykaryon as well as to their respective distances to each other.

## Rapid traffic of anterograde-directed cargo between Golgi in fused cells

After establishing our assay with Golgi resident enzymes, we evaluated transport of model anterograde soluble and membrane cargo proteins. We specifically chose model proteins whose aggregation state could be readily controlled, allowing us to test the cargo size dependence of the exchange mechanism. To control the size of a chimeric anterograde cargo, we took advantage of a well-established drug-dependent aggregation system (*Rivera et al., 2000*). We tested both a soluble cargo (hGH) and a transmembrane cargo (CD8$_{lumenal}$), each of which also contained four repeats of a self-aggregation domain (FM) and a fluorescent protein tag. The drug AP21998 maintains these proteins in the monomeric state; removal of AP21998 triggers reversible aggregation (*Volchuk et al., 2000*). Both the soluble and the membrane-attached version behave as bona fide anterograde cargo, moving rapidly from ER through Golgi to cell surface (*Volchuk et al., 2000*; *Lavieu et al., 2013*). However, the soluble and membrane-attached aggregates behave differently. Whereas soluble aggregates of FM4-hGH traverse the Golgi stack, aggregated CD8$_{lumenal}$ morphologically 'staples' Golgi cisternal membranes and remains static in the centers of Golgi cisternae (*Lavieu et al., 2013*).

Confocal video-microscopy of fused cells revealed that the green-labeled CD8$_{lumenal}$ was transported to the originally red-labeled acceptor Golgi in the presence of the AP21998 disaggregating drug (*Figure 3A*, Golgi 1), with an average of 12.9 ± 6.5 (n = 4) fold increase at 30 min post-fusion. Monomeric soluble cargo (hGH) also exchanged at similar rate (*Figure 3B*, Golgi 1). This showed that exchange is not limited to resident Golgi proteins and suggests that exchange of anterograde cargo can occur during the process of anterograde transport, at least in fused cells.

## Aggregation of cargo prevents inter-Golgi traffic

The removal of AP21998 to trigger aggregation of cargo in the Golgi completely abolished the inter-Golgi transport of the membrane-bound cargo CD8$_{lumenal}$ (*Figure 3A*, Golgi 2), with a 0.9 ± 0.3 (n = 3) fold increase at 30 min post-fusion. Similarly, aggregated soluble FM4-hGH was not exchanged efficiently with GT-GFP labeled acceptor Golgi (*Figure 3B*, Golgi 4).

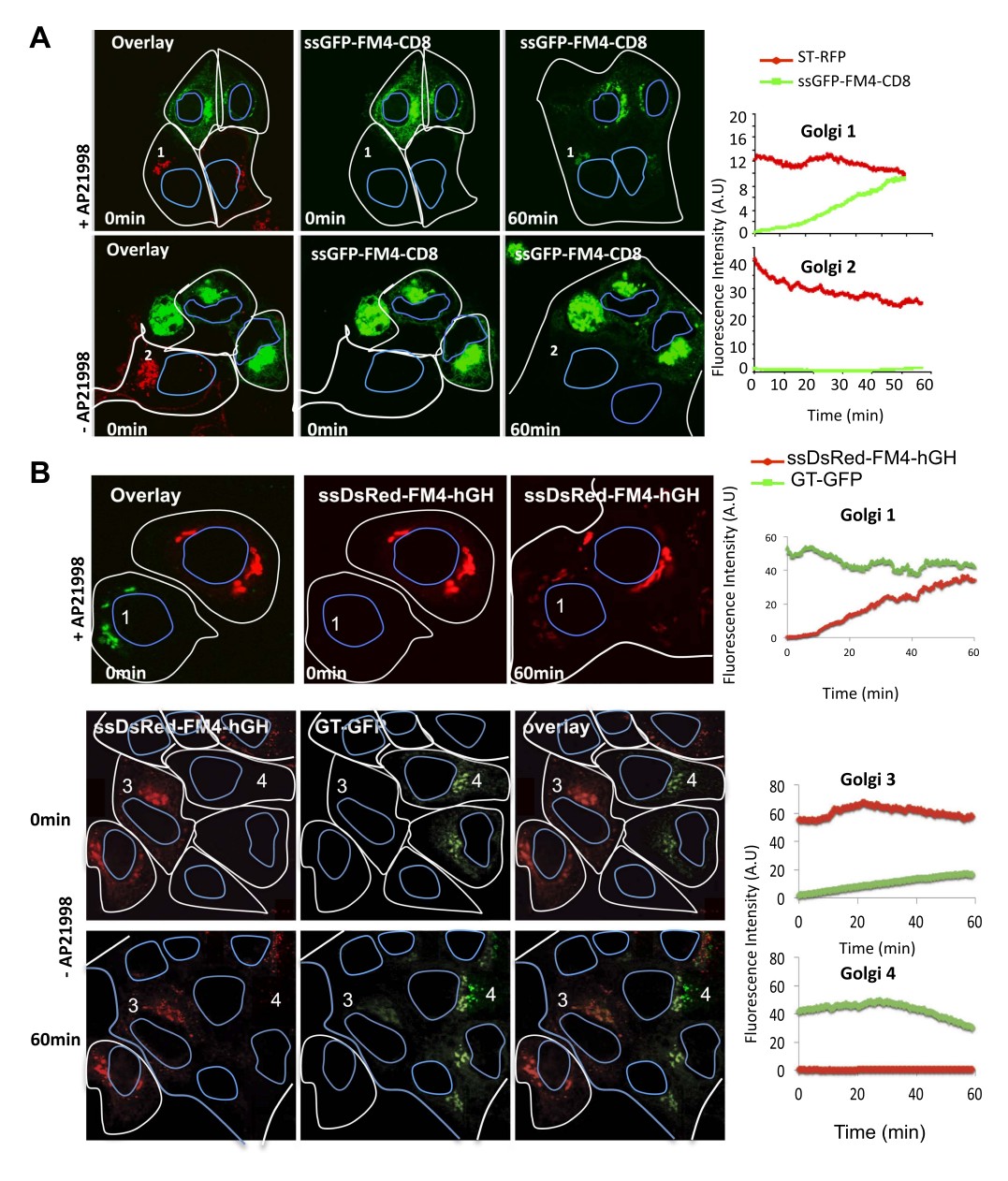

**Figure 3**. Inter-Golgi transport of small anterograde cargo. (**A**) HeLa cells expressing either ssGFP-FM4-CD8 or ST-RFP (+VSV-G) were mixed and fused. Before fusion, cells were incubated at 20°C for 2 hr in the presence of CHX (100 μg/ml) and AP21998 (500 nM) to trigger the release of the cargo from the ER and its accumulation in the Golgi. Both drugs were maintained during the imaging. For the study of the aggregated cargo (staples), AP21988 was removed 30 min before fusion, and cells were imaged in an AP21988-depleted medium at 20°C. Graphs show quantification of both markers over time for Golgi 1 and 2. Results are representative of three independent experiments. (**B**) HeLa cells expressing either ssDsred-FM4-hGH or GT-GFP (+VSV-G) were mixed and fused. As for (**A**) AP21988 was removed to trigger the formation of soluble aggregates within the Golgi. Graphs show quantification of both markers over time for Golgi 1 to 4. Results are representative of two independent experiments.

The following figure supplements are available for figure 3:

**Figure supplement 1**. Inter-Golgi transport is microtubule independent.

These results are important because they speak to the mechanism of transport between Golgi. Because re-aggregated CD8$_{lumenal}$ permanently resides in the Golgi membranes (*Lavieu et al., 2013*) and is not transported between Golgi areas, we can rule out that the traffic process somehow results from fragments of Golgi stacks (or entire mini-stacks) that detach by fission from one Golgi area and travel to another nucleus to join its Golgi area. The movement of Golgi mini-stacks over the required distances (approximately 5–50 μm) would be expected to be very slow without motor-based motility. We have extensively tested for the possible requirements of microtubules, which do not appear to be required (*Figure 3—figure supplement 1*).

Surprisingly, since they are rapidly transported within a single Golgi stack, large-soluble cargo aggregates that concentrate in the dilated rims of cisternae (*Volchuk et al., 2000*) 'are not' transported between separated Golgi (*Figure 3B*, lower panel). This suggests that the (unknown) carriers of small cargo are mobile and capable of traveling a long distance, whereas the 'carriers' of large aggregates of soluble cargo are less mobile.

## The ER is not a significant source of transferred protein

We performed all the experiments in the presence of the protein synthesis inhibitor cycloheximide to block production of new GFP-tagged cargo and to allow the ER to be drained of the vast majority of GFP-tagged cargo present before cells were fused. Nonetheless, in theory some of the cargo or resident protein transported to an exogenous Golgi could have originated from a small amount remaining in the ER, reflecting the established process of ER → Golgi rather than the novel process of Golgi → Golgi transport.

To place rigorous limits on this possibility we performed photo-bleaching experiments. In one approach, we measured the fluorescence recovery of photo-bleached Golgi within single or fused cells treated with H89, a kinase inhibitor that prevents ER export (*Aridor and Balch, 2000*). In single cells, we anticipate the absence of recovery of photo-bleached Golgi in presence of H89. Within fused cells, we reasoned that H89 should abolish the fluorescence recovery of photo-bleached Golgi only if the ER is the main source of cargo. As described above, one population of cells expressing the GFP-tagged anterograde cargo and a second population containing a RFP-tagged resident protein were mixed and fused in the presence of AP21998 (to maintain the FM-linked cargo in a monomeric state) to allow inter-Golgi exchange of both markers. 30 min post-fusion, the fused cells were treated with H89 for 15 min before and during the FRAP of Golgi of fused cells (*Figure 4*, Golgi 2). The average recovery at 30 min after-photo-bleaching was 34 ± 9.9% (n = 5). As a control, in the same population of cells that had not in fact fused (*Figure 4*, Golgi 1) no significant recovery was measured (7.2 ± 6.5%, [n = 4]), validating that ER → Golgi is abolished under H89 treatment. It is known that H89 also inhibits PKD (*Jamora et al., 1999*) and prevents exit from the TGN (*Muniz et al., 1996*). This broad spectrum of action of H89, which blocks ER → Golgi as well as TGN → PM, also suggests that inter-Golgi transport occurs exclusively between cis/medial/trans cisternae of separated Golgi.

Finally, to rule out any involvement of the ER, we tested if a Sar1 dominant mutant (Sar1H79G, a GTP-locked mutant) had any effect on inter-Golgi transport of disaggregated CD8. Although this mutant inhibits ER → Golgi transport and triggers the retention of Golgi resident enzymes within the ER (*Ward et al., 2001*), Golgi matrix proteins remain associated with membranes within the peri-nuclear area (*Ward et al., 2001*). This area could serve as an acceptor compartment during inter-Golgi exchange in the cell–cell fusion assay. We fused YFP-Sar1$_{H79G}$ with cells expressing disaggregated-DsRed-CD8 (+VSVG) accumulated in the Golgi (using the 20°C temperature block). DsRed-CD8, initially labeling the donor Golgi (Golgi 1, *Figure 4B*) progressively labeled the peri-nuclear area (presumably the Golgi remnant) of the acceptor cell that initially expressed the Sar1 mutant. The rate of the fluorescence increase in the acceptor cell was not significantly different from the one reported in the previous experiments performed in the absence of the Sar1 mutant (*Figure 3*), and was directly proportional to the decrease of the fluorescence at the donor Golgi, as expected if the donor Golgi is the only source of fluorescence within a freshly formed dikaryons. Note that the Sar1 mutant spreading into the dikaryons showed a peri-nuclear labeling reinforcement (*Figure 4B*, arrow) consistent with the localization (near the cis-Golgi/ERGIC) previously reported for Sar1 and GFP-Sar1H79 G (*Kuge et al., 1994*; *Venditti et al., 2012*).

We concluded that the contribution of ER (if any) during the inter-Golgi transport assay could only be minor.

Another possibility is that inter-Golgi transport of anterograde cargo is mediated via indirect transit through the endosomal pathway. To assess this possibility, we evaluated within fused cells the degree

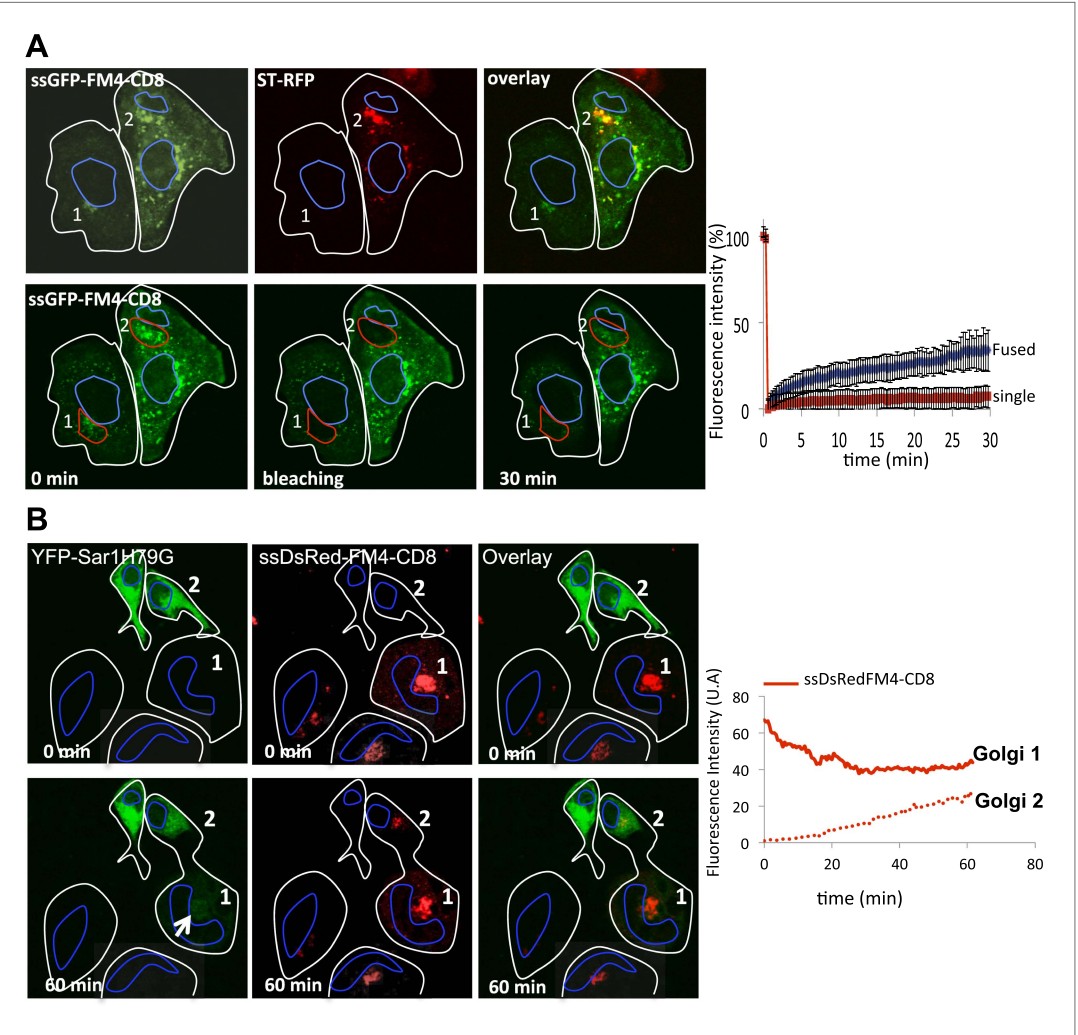

**Figure 4**. ER as a minor source of anterograde cargo during inter-Golgi transport. (**A**) FRAP performed on Golgi within fused or single cells treated with H89 (50 μM). Top panel, Golgi 1 (green) from a single cell and Golgi 2 (green and red) from fused cells 30 min post-fusion in presence of CHX and AP21988 at 20°C. Lower panel, 15 min after the H89 treatment (t = 0 min), Golgi 1 and 2 were photobleached and the fluorescence recovery was monitored by confocal video-microscopy. The graph shows the fluorescence intensity over time of the GFP marker within fused cells (di- or polykaryons) or single cells treated with H89. The values are the mean of three to five independent experiments. (**B**) Sar1H79G does not inhibit inter-Golgi transport of anterograde cargo. HeLa cells expressing either ssDsRed-FM4-CD8 (+VSVG) or YFP-SarH79 G were mixed and fused. Before fusion, cells were incubated at 20°C for 2 hr in the presence of CHX (100 μg/ml) and AP21998 (500 nM) to trigger the release of the cargo from the ER and its accumulation in the Golgi. Graphs show quantification of ssDsRed-FM4-CD8 over time at both donor (Golgi 1) and acceptor (Golgi 2) compartments.

The following figure supplements are available for figure 4:

**Figure supplement 1**. Inter-Golgi transport does not involved transit through endosomes.

of co-localization between the anterograde cargo and early/late endosomal markers. When cell expressing small GFP-CD8$_{lumenal}$ were fused for 15 min with cells expressing the early ensosomal marker Rab5A-RFP, no co-localization was observed by confocal microscopy, the two signals being clearly separated even when observed at high magnification (***Figure 4—figure supplement 1A***, red inset). Similarly, the late endosomal marker Rab7-GFP was clearly distinguishable from DsRed–CD8$_{lumenal}$. (***Figure 4—figure supplement 1B***). As a positive control, fused cells expressing DsRed–CD8 and GFP–CD8$_{lumenal}$ showed a strong co-localization of both signals (***Figure 4—figure supplement 1C***).

When quantified, the average Pearson's coefficient for the positive control was >0.8, whereas the coefficient for both endosomal markers samples were <0.4, synonymous of poor to no co-localization. These results suggest that endosomes are not involved in inter-Golgi trafficking.

## Transient transport intermediates pass between Golgi populations

In order to establish the existence of putative transport intermediates carrying markers between Golgi populations during the exchange process, we performed two types of photo-bleaching experiments in fused cells expressing disaggregated CD8$_{lumenal}$ accumulated in the Golgi.

In the first type of experiment, photo-bleaching served to lower the cytoplasmic background during exchange allowing the presumed intermediates to be imaged in transit between two Golgi areas. To simplify the analysis, we chose only heterokaryons containing only two cells including the (initially non-fluorescent) acceptor Golgi. At 30 min post-fusion, the entire volume of the heterokaryon was photo-bleached, sparing only its donor Golgi. Thus, the unbleached donor Golgi (*Figure 5A*, Golgi 1) is the only potential source of fluorescent cargo for trafficking to the acceptor Golgi. As exchange continues, the fluorescence of the donor Golgi should decrease and the fluorescence of the acceptor Golgi should correspondingly increase. During this process transport intermediates carrying the fluorescent cargo between the two Golgi, if such exist, should in principle be present in between the two Golgi areas.

30 min after the photo-bleaching, a 23.9 ± 6.4% (n = 3) fluorescence recovery of the pre-bleaching fluorescence was observed at the acceptor Golgi (consistent with the 34% previously measured in the H89-treated fused cells) accompanied by a 24.8 ± 9.03% (n = 3) decrease in fluorescence of the donor Golgi. This independently confirms that the donor Golgi is a sufficient source for inter-Golgi exchange and established the pre-conditions to look for transport intermediates.

In fact, during the exchange period we readily observed numerous fluorescent "dots" present in the intervening cytoplasm (*Figure 5A*, gray arrows). We estimate that there may be as many as ~20,000 of such dots per heterokaryon over 30 min considering that we monitored ~6 dots/frame (with 1 frame/5 s over 30 min, in a 500 nm focal plane within HeLa cells that are ~5 µm thick). These particles were not easily distinguishable above background fluorescence when the inter-Golgi assay was performed without prior photo-bleaching, probably because the background signal was is too high.

To further test if fluorescently-labeled cargo present in the dots is a bona fide intermediate that contain cargo in transit and if they carry sufficient fluorescent cargo to account for the bulk of exchanged cargo, we performed a sequential 'double FRAP' experiment. This was performed by spinning-disk microscopy to track all the fluorescent objects within the entire volume of a dikaryon. First, 30 min post-fusion, we photo-bleached the entire volume of the fused cell (including the ER and the diffusing fluorescent dots [*Figure 5B* panel 2]) except the two Golgi, which remained the only source of fluorescence. As described above, fluorescent dots that emanated from the Golgi rapidly appeared, increased in number and spread throughout the cytoplasm, resulting in an increase of the fluorescence intensity measured in the inter-Golgi space (*Figure 5B* panel 3, Graph). Then, after a 30 min recovery period, a second photo-bleaching was performed on the two Golgi (panel 4), the only source of fluorescence being now exclusively provided by the freshly formed cytoplasmic dots. We reasoned that if these fluorescent dots are truly the transport intermediates, then they should be able to repopulate the photo-bleached Golgi. After 30 min recovery, the gain of fluorescence observed at the Golgi (+56 ± 14% of the total fluorescence initially emanating from the dots before the second photo-bleaching) was very similar to the loss (−63 ± 14%) of the fluorescence intensity monitored in the inter-Golgi space (and emanating from the dots) (Graph, *Figure 5B*). The subtle difference (around 7%) between the loss (in the Golgi inter-space) of and the gain (at the Golgi) of fluorescence that suggests a loss of fluorescent material during the experiment, could be fairly attributed to moderate photo-bleaching that usually occurs during the time course of our imaging. Note that in these experiments, ST-RFP was coexpressed with GFP–CD8$_{lumenal,}$ and allowed for rigorous identification of the Golgi area and to measure the GFP fluorescence recovery (*Figure 5B*, red squares).

These results strongly suggest that the Golgi-derived small fluorescent dots are the diffusible transport intermediates responsible for the inter-Golgi transport.

## Inter-Golgi transport is ARF1 and ε–COP dependent

One possibility is that the small diffusible transport intermediates, which we visualize by confocal microscopy as fluorescent dots, are COPI vesicles, which are known to copiously bud from Golgi

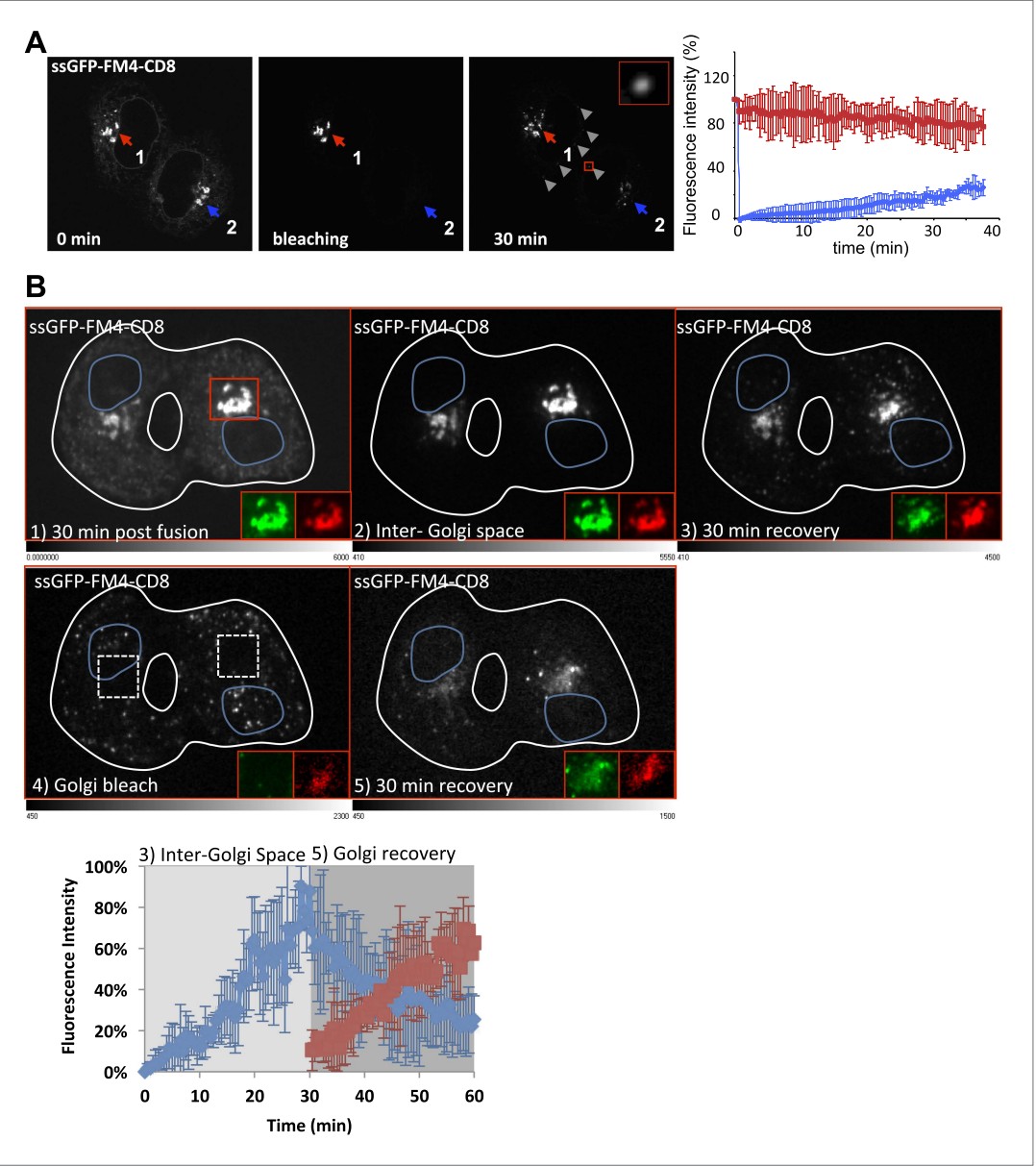

**Figure 5**. Diffusible inter-Golgi transport intermediates. (**A**) 30 min post fusion, the entire volume a dikaryon (ER and acceptor Golgi 2) was photobleached except for the Golgi 1 that remains the only source of fluorescence. Small diffusing fluorescent dots (gray arrow heads) were observed throughout the cytoplasm of fused cells. The graph shows the intensity of both Golgi (acceptor in blue and donor in red). Values are the mean of three independent experiments. (**B**) Cells expressing ssGFP-FM4-CD8 and ST-RFP (+VSV-G) were incubated at 20°C for 2 hr in the presence of CHX (100 µg/ml) and AP21998 (500 nM) to trigger the release of the cargo from the ER and its accumulation in the Golgi. Both drugs were maintained during the imaging. (1) Cells were fused and imaged 30 min post fusion. (2) Entire cell volume (Inter-Golgi space) was bleached, except the two Golgi and was allowed to (3) recover for 30 min. (4) Donor and acceptor Golgis were bleached (dashed white boxes) and allowed to (5) recover for 30 min. Red box indicates zoom area (ssGFP-FM4-CD8 in green and ST-RFP in red). Images are representative of three independent experiments. Images displayed are single slices of z-stacks (21 one µm slices) acquired. LUTs displayed below each grayscale image. Graph, quantification of total fluorescence intensity in region outside of the Golgis (blue line) and in Golgi areas (red line). These values represent the mean of three independent experiments. Fluorescence intensity for each timepoint was calculated over entire z-stack.

The following figure supplements are available for figure 5:

**Figure supplement 1**. ARF1$_{Q71L}$ abolished fluorescence recovery in the inter-Golgi zone and at the Golgi.

membranes and contain both anterograde and retrograde cargoes. The small GTPase ARF1 promotes vesicle budding from various donor membranes, including Golgi membranes (*Serafini et al., 1991a*; *Bremser et al., 1999*). To test the possible requirement of ARF1 in inter-Golgi transport, we introduced into the fused cell a dominant-interfering mutant of ARF1 (ARF1$_{Q71L}$) that can bind but not hydrolyze GTP, thereby accumulating coated COPI vesicles (*Tanigawa et al., 1993*). To do this we included a third cell population expressing the DsRed–ARF1$_{Q71L}$ mutant, with cells co-expressing VSV-G and GT-CFP and cells expressing GT-YFP.

After this three-way fusion, confocal video-microcopy revealed that ARF1$_{Q71L}$–DsRed spread effectively throughout the polykaryons and localized to exogenous Golgi membranes (*Figure 6A*, Golgi 1 and 2). However, inter-Golgi exchange of yellow and cyan-labeled GT no longer took place. Each marker remained within its parental Golgi even 1 hr after fusion. This suggests that the GTPase activity of ARF1 is required for inter-Golgi exchange of resident Golgi proteins.

To test the possible ARF1 dependence of inter-Golgi transport of anterograde cargo, we mixed and fused two cell populations, one expressing the disaggregated GFP–CD8$_{lumenal}$ cargo within their Golgi, the other expressing DsRed–ARF1$_{Q71L}$. Once again, the ARF1 mutant diffused throughout the polykaryon and bound to the exogenous Golgi membranes (*Figure 6*, Golgi 1) and prevented inter-Golgi transport of this small anterograde cargo, which remained in its original Golgi. Remarkably, when the ARF1 mutant is replaced with ARF1WT, inter-Golgi transport of the cargo is unaltered (*Figure 6—figure supplement 1*), demonstrating that the transport inhibition was specific to ARF1 mutant.

To test if the dots (transport intermediates) are ARF GTPase-dependent, we performed sequential double FRAP experiments in a heterokaryon resulting from the fusion between cells containing disaggregated GFP–CD8$_{lumenal}$ cargo accumulated in the Golgi and cells that contained DsRed–ARF1$_{Q71L}$ (*Figure 5—figure supplement 1*). As shown above (*Figure 6*), GFP–CD8 remained at the donor Golgi 30 min–post fusion, whereas the ARF1 mutant decorated each Golgi of the polykaryon. After the first photo-bleaching, virtually none of the GFP-fluorescent dots that were normally observed (*Figure 5B*) could be visualized when ARF1-GTP locked mutant was present (*Figure 5—figure supplement 1*). Consistent with this lack of repopulation of transport intermediates in the inter-Golgi zone, the GFP-fluorescence recovery in that zone was less than 5% of the recovery measured in polykaryons lacking the ARF1 mutant (*Figure 5B* and *Figure 5—figure supplement 1*). As expected, and consistent with the absence of production of transport intermediates, no significant GFP-fluorescence recovery was observed after the second round of photo-bleaching at the Golgi area (*Figure 5—figure supplement 1*). Because of the lack of resolution, we do not know if in the presence of the ARF1-locked mutant, transport intermediates were actually produced but remained in the vicinity of the donor Golgi in a coated form, or if the ARF1 mutant prevented complete formation of COPI vesicles. In any case, this result suggests that the GTPase activity of ARF1 involved in the formation of COPI vesicles is required for generation of diffusible transport intermediates that are responsible for inter-Golgi exchange.

To test the possible requirement for coatomer, we examined inter-Golgi exchange of Golgi resident proteins (ST-RFP and GT-GFP) at the restrictive temperature of 39°C within fused wild-type CHO cells (WT) or Ldl-F mutant cells, a well characterized mutant that is thermo-sensitive for the ε−COP subunit of the coatomer (*Guo et al., 1994*). This process could not be monitored in real time at this temperature, because our microscope was not equipped with an appropriate temperature-controlled chamber. Therefore, cells were fixed 15 min and 60 min after fusion to determine if inter-Golgi exchange had occurred. To estimate the degree of colocalization of the two Golgi markers, we calculated the Pearson's coefficient for each picture (*Figure 6C*). At 60 min post-fusion, the average Pearson's coefficient in fused WT cells was 0.69 ± 0.11 (n = 30), indicative of strong co-localization. In contrast, in fused mutant Ldl-F cells, the negative value of the Pearson's coefficient (−0.29 ± 0.09 [n = 30]) illustrated the absence of significant co-localization, indicating that ε-COP is required for the inter-Golgi transport of resident proteins.

All the previous experiments with the anterograde cargo were performed using a 20°C temperature block to retain the cargo within the Golgi. Unfortunately, this is not possible in the mutant CHO cells if we wish to retain the restriction on coatomer function to test for a requirement. When analogous experiments were performed with the CHO cells to assess the requirement of the coatomer, the 39°C temperature shift of course triggers rapid exit of the anterograde cargo from the Golgi to the plasma membrane. Therefore, we could not perform a suitable control to rigorously validate the absence of inter-Golgi transport of anterograde cargo.

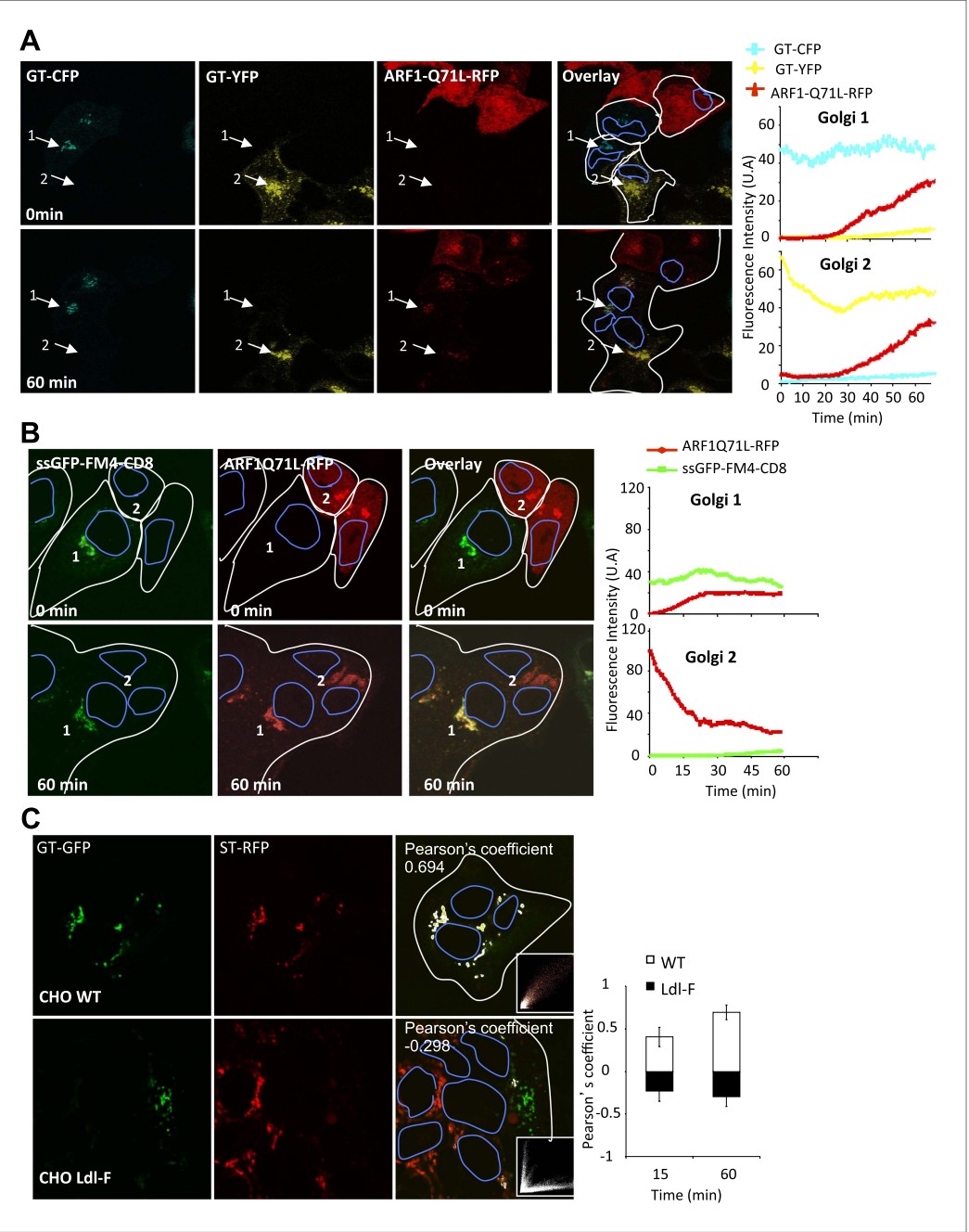

**Figure 6**. Role of ARF1 and ε–COP on inter-Golgi transport. (**A**) Mixed and fused HeLa cells expressing GT-CFP (+VSV-G), GT-YFP (+VSV-G) or ARF1$_{Q71L}$–DsRed, in presence of CHX. Graphs show the fluorescence intensity over time for each fluorescent marker within Golgi 1 and 2. Results are representative of two independent experiments. (**B**) Mixed and fused HeLa cells expressing either ssGFP-FM4-CD8 (+VSV-G) or ARF1$_{Q71L}$–DsRed in the presence of AP21988 and CHX at 20°C. The graphs show the fluorescence intensity overtime of each marker for each Golgi. Results are representative of three different experiments. (**C**) Mixed and fused WT-CHO cells or LdlF-CHO cells expressing GT-GFP or ST-RFP (+VSV-G), in presence of CHX. After fusion, cells were incubated at 39°C for 15 or 60 min, fixed, and monitored by confocal microscopy. Graph shows the relative Pearson's correlation coefficients for each cells type at each time point. Values are the mean of three independent experiments, with 10–15 polykaryons being analyzed for each condition. Pictures are representative of three experiments and illustrate cells at 1-hr post fusion.

The following figure supplements are available for figure 6:

**Figure supplement 1**. ARF1 WT does not prevent inter Golgi exchange of small anterograde cargo.

Our results show that inter-Golgi transport requires both ARF1 and COPI, suggesting that COPI vesicles are the transport intermediates.

## Inter-Golgi transport intermediates are the size of COPI vesicles and contain coatomer

To further test the possibility that the putative carriers are COPI vesicles, we used STED microscopy to pinpoint their size and their content at ~80 nm resolution (*Pellett et al., 2011*). In these experiments we used a primary monoclonal antibody against β-COP and a secondary antibody coupled to the STED compatible dye, ATTO647, to identify the putative COPI coated carrier. We fused donor cells that expressed both a tagged resident protein (ST-RFP) and small anterograde membrane cargo (GFP–CD8$_{lumenal}$ in presence of AP21998) with untransfected acceptor cells. 15 min after fusion, when the markers were in mid-transit to the acceptor Golgi, we fixed the samples and performed immunofluorescence against β-COP to label COPI containing membranes. This results in labeling of the Golgi itself and the dots/putative carriers (*Figure 7* and *Figure 7—figure supplement 2*).

As a standard for comparison with the fluorescent dots found between Golgi, we measured the size of in vitro COPI vesicles by STED microscopy. We determined their mean diameter (*Figure 7A*) to be 114 ± 11 nm n = 64, consistent with the outer diameter reported from electron microscopy (*Orci et al., 1986*). We then measured the size of the β-COP positive transport intermediates (dots) within fused cells by STED microscopy (*Figure 7B*). The mean size of the dots containing COPI was 126 ± 15 nm, n = 181, and approximately the same size as the in vitro vesicles. Note that the vast majority (around 90%) of the dots containing either the anterograde cargo or the Golgi resident enzymes were negative for the COPI staining, consistent with the predictions of the vesicular model. Interestingly these uncoated structures showed a slightly reduced diameter 105 ± 13 nm, n=60, which may reflect the absence of coatomer. Analysis of the shape (defined as the length of the major axis/length of the minor axis) of the transport intermediates and the in vitro carriers revealed similar radial symmetry (*Figure 7—figure supplement 2*). However, according to our 3D modeling, both the in vivo putative transport intermediates and the in vitro COPI vesicles were not as symmetrical as modeled vesicles (*Figure 7—figure supplement 2*). This could be attributed to variation generated by the length and the spatial orientation of the antibodies (both primary and secondary) that bound heterogeneously to the coatomer. This subtle asymmetry fit with a recent high-resolution 3D study of in vitro COPI vesicles, which showed variable shapes of the vesicles, as well as a non-homogenous surface distribution of the coatomer (*Faini et al., 2012*). Finally, our shape analysis and 3D modeling clearly establish that the COPI positive transport intermediates are not long tubules (30 nm × 300 nm), which show a significantly higher degree of asymmetry. However, it is impossible to rule out from such modeling that the carrier is a hypothetical, partially COPI coated small tubule. With a resolution of 80 nm, a 30 nm diameter tubule that is 150 nm long is indistinguishable from a 110 nm vesicle (as confirmed by simulations of STED images of vesicles and tubules; *Figure 7—figure supplement 2*).

Are the anterograde cargo and the resident protein cargo present in the same or different COPI vesicles? Quantification using size filtering based on the in vitro COPI vesicles results (size < 150 nm) showed the following heterogeneous distribution when GFP–CD8$_{lumenal}$ was co-expressed with ST-RFP (*Figure 7B*): 66 ± 20% of the COPI-positive objects carried the anterograde cargo alone, 20 ± 11% carried the retrograde cargo alone and 15 ± 8 % carried both (*Figure 7B*), consistent with the three populations observed by confocal microscopy (*Figure 7—figure supplement 1*). When the same anterograde cargoes tagged with either green or red fluorophores were co-expressed, 87 ± 3% of the objects carried both cargoes, as expected for a co-localization positive control. None of the dots (again, containing one or both markers) could be stained with an antibody to the COPII coat subunit SEC31 (*Figure 7—figure supplement 1*), yet again ruling out again any involvement of ER export.

Our inter-Golgi transport assay has two main limitations.

First, all the experiments presented in this study were performed at 20°C, and one could argue that this non-physiological temperature could create missorting because of accumulation of over-expressed cargo at the TGN. This so-called 20°C block has never been reported to prevent intra-Golgi trafficking (neither anterograde or retrograde). Instead it has been shown that 20°C slows-down the trafficking, and results in retaining a large portion of the anterograde cargo within the trans-Golgi, with a significant amount remaining in the upstream cisternae that could serve as a back-up for inter-transport (*Van Deurs et al., 1988*). However, to rule out that our results were reflecting non-specific sorting of the anterograde cargo at the TGN, we co-expressed the anterograde cargo with a dominant interfering

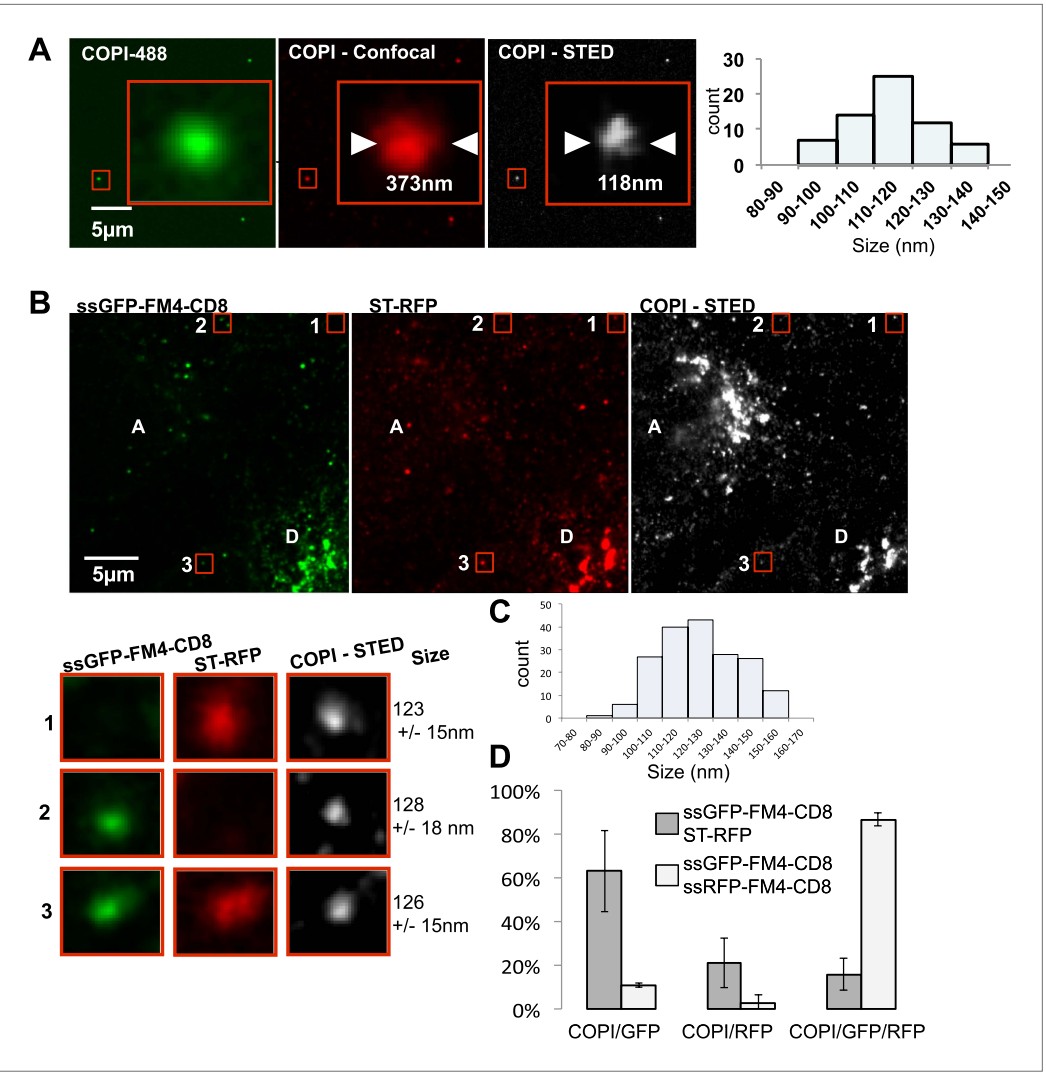

**Figure 7**. Inter-Golgi transport intermediates are compatible with COPI vesicles. (**A**) In vitro prepared COPI vesicles labeled with Alexa488 were attached to glass coverslips. Immunofluorescence against COPI was performed, and samples were imaged by both confocal and STED microscopy. The size of COPI vesicles was determined by STED microscopy using a custom Matlab routine by fitting to a 2D Lorenztian function and confocal images were fit to a 2D Gaussian function. Graph, size distribution of 64 in vitro COPI vesicles fit with a 2D Lorentzian function. The mean is 114 nm. (**B**) HeLa cells expressing ssGFP-FM4-CD8, ST-RFP and VSV-G were incubated at 20°C in the presence of AP21998 and CHX for 2 hr, fused, and fixed after 30 min. Immunofluorescence was performed against β-COP to label all the Golgi related structures. COPI like transport intermediates imaged by STED microscopy were identified based on size and the contents were evaluated. Numbered boxes illustrating types of cargo containing COPI like intermediates. Red squares, different objects at low and high magnification. The sizes of the intermediates were determined using a 2D Lorentzian function. Graph (**C**) shows the size distribution of 179 carriers fit with a 2D Lorentzian function, The mean is 128 nm. Graph (**D**) shows the distribution of COPI spots positive for either ssGFP-FM4-CD8 cargo or ST-RFP, or both (gray columns). As a positive control we used cells co-expressing ssRFP-FM4-CD8 instead of ST-RFP (white columns). Control: 48 cells, 138 spots analyzed; Experimental: 31 cells, 179 spots analyzed. Results are representative of two independent experiments, each one being duplicated.

The following figure supplements are available for figure 7:

**Figure supplement 1**. Inter-Golgi carriers are COPI positive and COPII negative.

**Figure supplement 2**. Discriminating tubules from vesicles with STED.

mutant of the protein-Kinase D (PKD(DI)), known to block the TGN exit of anterograde cargo (*Baron and Malhotra, 2002*; *Figure 8C*). Cells expressing CD8-GFP and PKD(DI) were pre-incubated at 37°C for 15 min in the presence of AP91988 and CHX, to allow for Golgi targeting of the cargo. As expected, we did not observe any leak to the PM. These cells were fused with other cells expressing DsRed-tagged resident enzyme and PKD(DI), and the inter-Golgi transport was recorded at 32°C for 1 hr. The anterograde cargo was efficiently transported to the acceptor Golgi (*Figure 8A*, Golgi 1 and Graph) that originally contained only the resident enzyme. Note that 1 hr post-fusion, cells were fixed and processed for immunofluorescence against GST-PKD(DI) to rigorously assess for the presence of the PKD mutant within the very same polykarion (*Figure 8B*). This result suggests that inter-Golgi transport is not due to a missorting of over-expressed cargo at the TGN.

The second limitation is the over-expression of the cargoes. It is possible that we artificially increased the probability of loading the highly concentrated cargo within the carriers. For instance, the excess of anterograde cargo localized in the trans-Golgi may now be transported to the cis-Golgi of the acceptor Golgi through a retrograde pathway. On the other hand, over-expression of the resident enzymes may alter their proper localization, resulting now in an anterograde transport of these resident proteins when they are in large excess. We cannot rule out that these phenomena happened to some extent in our assay. However, if these assumptions are correct, the two types of cargo should always be carried within the same transport intermediates when the cargoes are released from the same donor Golgi. Our results clearly establish that this in not the case (*Figure 7B*). Another interpretation would be to consider that the inter-Golgi transport intermediates would be COPI vesicles, which contain exclusively anterograde cargo in ~60% of the cases. ~20% would be COPI vesicles containing only Golgi resident enzymes that are retrograde-directed, needed in any model to maintain the steady-state distribution of resident Golgi enzymes in a dynamic equilibrium. It is tempting to suggest that the ~20% of COPI vesicles containing both anterograde cargo and resident enzymes would be anterograde-directed, contributing to the steady-state distribution of resident enzymes by forward flow. As is often the case, the truth may lie in between these two most extreme interpretations.

## Discussion

This report describes a study specifically intended to test the predictions of the vesicular transport model. Our reasoning was that only mobile carriers, such as COPI vesicles, should be able to travel the long distance between well-separated Golgi of fused cells. Our results clearly establish that only small cargoes (anterograde or resident proteins) are exchanged, showing that inter-Golgi carriers are capable of size-filtration, and that small cargo transport must utilize a distributive mechanism different from the mechanism used by large soluble aggregates. More than half of the carriers exclusively contained anterograde cargo, suggesting sub-populations of COPI vesicles as previously documented by immuno-EM (*Orci et al., 1997*).

Because the imaged carriers contained coatomer, are indistinguishable in size from authentic COPI vesicles, and require both coatomer and ARF1 to function, the simplest interpretation is that the bulk of inter-Golgi transport of small cargo is mediated by COPI vesicles. Certainly, the double photo-bleaching experiments show that the bulk of the cargo flux between Golgi is mediated by fluorescent 'dots' that function as carriers. However, we cannot strictly rule out that the genetic effects of ARF1 and ε−COP mutations on inter-Golgi transport are somehow an indirect coincidence or that the carrier is some kind of very short tubule that is partially coated and capable of fission (at one Golgi) and fusion (at another). Correlative electron microscope imaging of the fluorescent dots, including immunolabeling of coatomer, would theoretically be needed to address this distinction, which is probably semantic in any case since many COPI vesicles have uncoated portions (*Faini et al., 2012*). Because the dots/COPI vesicles are highly dilute in cytoplasm, the rarity of finding such vesicles in single EM sections together with the inherent inefficiency of immunolabeling at the EM level, makes such an effort quixotic at best. Although we cannot completely rule out the possibility that exogenous glycosyltransferases exceed the capacity of the Golgi retention machinery and thus begin to behave like soluble cargoes, the relative abundance of COPI containing carriers harboring Golgi resident glycosyltransferase (~40% of the total) suggests that these proteins are engaged in a dynamic equilibrium of anterograde and retrograde flow to maintain their differential steady-state cis-trans distributions. Their presence in transport vesicles is a required feature of the mobile cisterna model, as incorporated in the concept of cisternal 'maturation' (*Glick and Malhotra, 1998*), but may also be important for establishing and maintaining their cis-trans distributions by bi-directional transport (*Pelham and Rothman, 2000*) among static cisternae. We

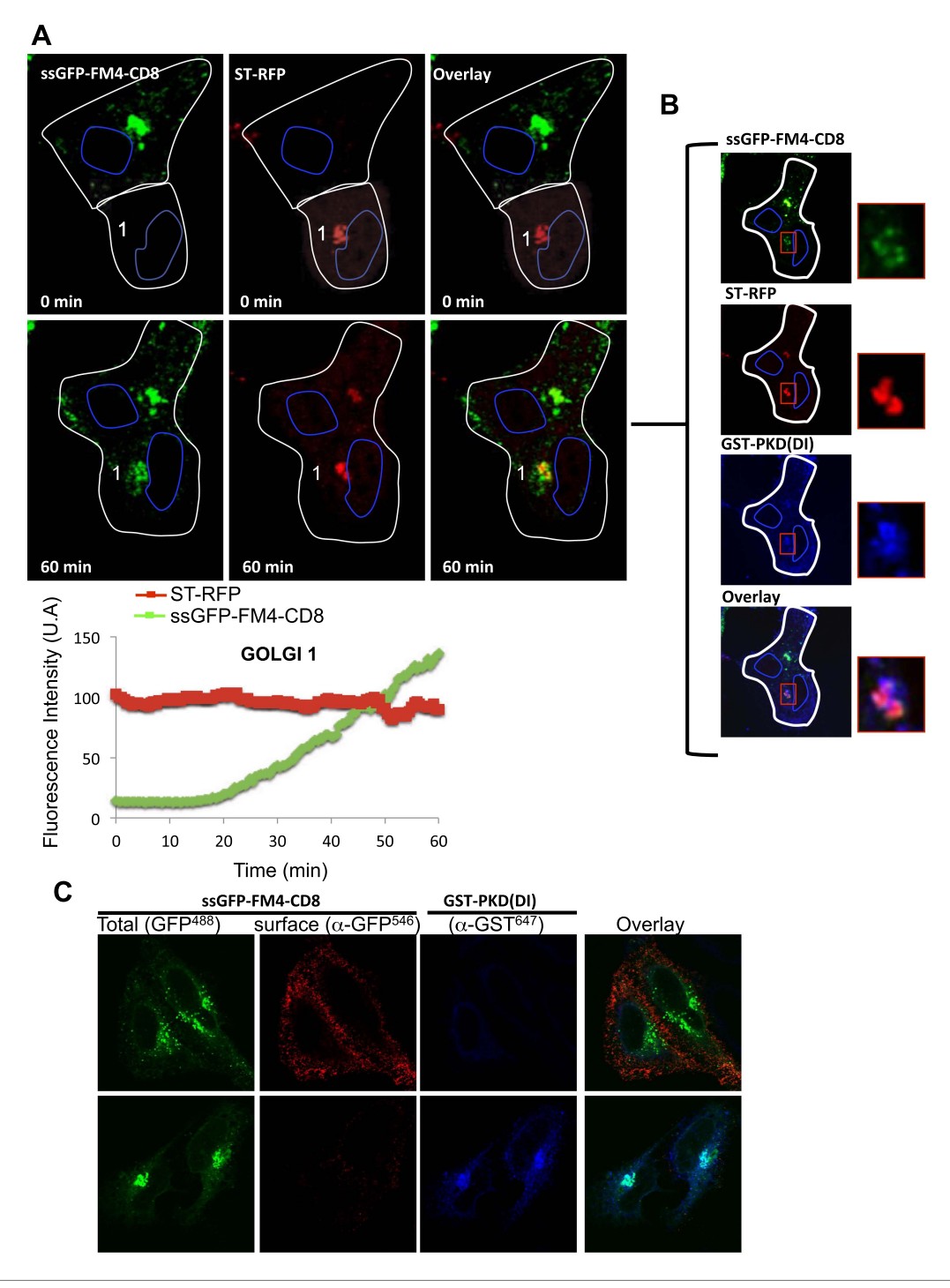

**Figure 8**. Inter-Golgi exchange of anterograde cargo at 37°C. (**A**) HeLa cells expressing ss-GFP-FM4-CD8 and GST-PKD(DI) were mixed and fused with cells expressing ST-RFP, GST-PKD(DI) and VSV-G. Prior to fusion, cells were incubated at 37°C for 15 min in the presence of the disaggregating drug and CHX. Cells were monitored by video confocal microscopy at 32°C in the presence of the disaggregating drug and CHX. Graphs show quantification of both markers over time for Golgi 1. Results are representative of two independent experiments. (**B**) 1 hr post-fusion, cells were fixed and prepared for immunofluorescence against GST to assess the presence of the PKD(DI). Note that the confocal micrograph showed the very same field of cells (shown in **A**) after fixation. (**C**) PKD(DI) inhibits plasma membrane targeting of ss-GFP-FM4-CD8. HeLa cells expressing ss-GFP-FM4-CD8 alone (upper panel) or

*Figure 8. Continued on next page*

*Figure 8. Continued*

with GST-PKD(DI) (lower panel) were incubated for 1 hr at 37°C in the presence of the disaggregating drug. Non-permeabilized cells were incubated on ice and incubated with an anti-GFP antibody (detected with a secondary antibody labeled with Alexa-546) to assess for cell surface exposure of ss-GFP-FM4-CD8. Then, cells were fixed, permeabilized and processed for immunofluorescecne against GST (using a secondary antibody labeled with Atto-647).

note that previous reports on the prevalence of glycosyl-transferases in COPI vesicles have yielded variable results (*Orci et al., 2000*; *Martinez-Menarguez et al., 2001*; *Cosson et al., 2002*).

The present results naturally suggest that transport of small cargo is bi-directional and mediated by COPI-coated vesicles. The data suggest that COPI vesicles traffic both secretory cargo and steady-state Golgi resident enzymes among stacked cisternae that are stationary over many hours in non-dividing cells. Further, they suggest that large soluble cargo are transported across the Golgi by a specialized mechanism that involves carriers which remain closely linked to the stack as the progress across it. While these inferences are as strong as current imaging technology permits, super-resolution methods are rapidly advancing and it will be important to further test these models as it becomes possible to visualize the content and movements of individual carriers in real time in living cells.

## Materials and methods

### Plasmids

GT-GFP and ST-RFP were a gift from J Roher (*Schaub et al., 2006*). The VSV-G-encoding vector was provided by JK Rose (*Florkiewicz and Rose, 1984*). ss-GFP-FM4-CD8 (CD8$_{lumenal}$)was generated by sequential insertion of GFP and CD8 encoding sequences into a pC4-FM4 backbone vector (ARIAD), using XbaI/SpeI compatibility, and BamH1 restriction site. CD8-GFP was amplified from pC4-CD8-GFP as previously described (*Lavieu et al., 2010*). ss-DsRed-FM4-hGH was created by adding DsRed into pC4S1-FM4-FCS-hGH (ARIAD) using XbaI/SpeI compatibility. ARF1$_{Q71L}$ was cleaved from ARF1$_{Q71L}$–GFP, a gift from G Romero (*Vasudevan et al., 1998*) and inserted into a mDsRed vector (Clontech). GST tagged PKD-(DI) was a gift from Vivek Malhotra. GT-CFP and GT-YFP and YFP-Sar1H79 G (*Quintero et al., 2010*) were a gift from C Giraudo. RAb5A-RFP and RAb7-GFP were a gift from P De Camilli.

### Cell culture and transfection

HeLa cells were maintained at 37°C in 5% CO$_2$ in DMEM (Gibco, Grand Island, NY) supplemented with 10% FBS (Gibco). CHO cells were maintained at 34°C in 5% CO2 in F-12 Glutamax (Gibco) supplemented with 10% FBS. Cells were transfected using lipofectamine 2000 (Invitrogen, Grand Island, NY) as recommended by the manufacturer. When cells were co-transfected (VSV-G/other fluorescent-tagged protein), we used a 3/1 ratio.

### Cell fusion

Cells were plated at $6 \times 10^4$ (HeLa) or $1 \times 10^5$ (CHO) cells/well in 24-well plates. Cells were transfected with the plasmids of interest for 6 hr. Transfection media was removed, and cells were washed with complete culture media prior to being detached (trypsin 0.05% 2 min) and mixed overnight in glass bottom microwell dishes (HeLa cells), or on coverslips coated with collagen type I (CHO cells).

2 hr before fusion, HeLa cells were incubated at 20°C in HBSS (Gibco) supplemented with 10% FBS and 100 μg/ml CHX (Sigma, St. Louis, MO) (imaging media). CHO cells were kept in F12 media containing 10% FBS and 100 μg/ml CHX. For live imaging, HeLa cells were incubated on the stage of the microscope for 1 min in 37°C pre-warmed fusion buffer (10 mM Na$_2$HPO$_4$, 10 mM NaH$_2$PO$_4$, 150 mM NaCl, 10 mM 2-(*N*morpholino) ethanesulfonic acid [MES], 10 mM *N*-2-hydroxyethylpiperazine-*N*9-2-ethanesulfonic acid [HEPES], pH 5). Subsequently, the fusion buffer was replaced by the imaging media. CHO cells were incubated 1 min at 37°C in the fusion buffer before being incubated at 39°C with pre-warmed complete F12 media. Unless mentioned otherwise, all the experiments were performed in the presence of 100 μg/ml CHX at 20°C.

### Immunofluorescence and confocal microscopy

HeLa cells expressing CD8-GFP-FM4, ST-RFP, and VSV-G were fused with untransfected cells. Cells were fixed with PFA 4%, permeabilized with 0.1% Triton X100, and incubated with an anti-β–COP antibody (1/1000, EAGE clone, rabbit) or an anti-hsec31 antibody (1/1000, rabbit; *Shugrue et al.,*

*1999*), and an anti-rabbit Alexa-633 conjugated secondary antibody (1/2000; Invitrogen). GST antibody was purchase from Abcam, Cambridge, MA. Cells were then processed for confocal microscopy imaging, which was performed in multi-tracking mode on either a Zeiss LSM510 or a Zeiss LSM510 META. Images were analyzed using Zeiss LSM510 software or using ImageJ (co-localization finder plugin).

## Immunofluorescence and STED microscopy

Cells were fixed with 4% paraformaldehyde for 15 min at room temperature, permeabilized for 3 min with 0.3% NP40 and 0.05% Triton-X, blocked with 10% normal goat serum for 1 hr at room temperature and incubated with primary antibodies overnight at 4°C. Cells were incubated with secondary antibodies for 1 hr at room temperature and mounted in mowiol. All antibodies were used at a 1:1000 dilution: α-COPI, CM1A10, α-RFP rabbit polyclonal (Invitrogen); α-mouse ATTO647N (Active Motif, Carlsbad, CA) and α-rabbit Alexa543 (Invitrogen). STED images were obtained using a commercial Leica TCS STED microscope (*Pellett et al., 2011*).

## Size measurement method

The size of in vitro COPI vesicles were measured using a custom MATLAB routine by fitting STED images to a 2D Lorenztian function and confocal data to a 2D Gaussian function.

## Co-localization method

For each channel, 100 background regions of interest were randomly selected A normal distribution of the brightest pixel in each ROI was created and the mean and standard deviation was calculated. A thresholding value of the mean plus one standard deviation was used as a cutoff. Using Volocity, COPI positive transport intermediates were identified based on the size data from in vitro generated COPI vesicles. Intermediates identified were evaluated for the presence of GFP or RFP signal above the calculated thresholding value. Only intermediates positive for another marker were included in the STED image simulations.

We simulated vesicles of 110-nm diameter and tubules of 30 nm diameter and 80, 110, 150 and 300 nm lengths on a three-dimensional grid of $200^3$ cubic voxels of 5 nm size with the software Inspector (written by Dr Andreas Schoenle, Max Planck Institute for Biophysical Chemistry, Goettingen). All structures were assumed to be surface-labeled, that is to carry signal only on a shell defined by the delimiters of the structures ±5 nm. The tubules were simulated in a set of different orientations to represent an isotropic distribution. Given that the size of the organelles is significantly smaller than the axial resolution of our STED microscope (maximum 300 nm vs ~700 nm), the rotated structures were projected into the focal plane before convolving them with a radially symmetric Lorentzian-shaped STED PSF with 80 nm full-width at half maximum which matches to the experimental performance of the used STED microscope. Poisson-distributed background and signal shot noise was applied and adjusted to experimentally observed signal-to-noise levels creating 150 structures of each shape. This data was analyzed with the same custom-written MATLAB routine used for the experimental data analysis.

## Acknowledgements

JER thanks Prof Suzanne Pfeffer (Stanford University) for suggesting the idea of re-investigating inter-Golgi transfer in fused cells with modern technology. We thank ARIAD pharmaceuticals for providing us with the secretion/aggregation kit reagents. Thanks to Pietro De Camilli, Claudio Giraudo, Monty Krieger, Jack Rohrer, Guillermo Romero, John K Rose, and Fred Gorelick for sharing their reagents. We thank Felix Wieland and his lab members who provided purified COPI vesicles.

## Additional information

### Funding

| Funder | Author |
| --- | --- |
| National Science Foundation | Patrina A Pellett |

The funder had no role in study design, data collection and interpretation, or the decision to submit the work for publication.

## Author contributions

PAP, Acquisition of data, Analysis and interpretation of data, Drafting or revising the article, Contributed unpublished essential data or reagents; FD, Acquisition of data, Contributed unpublished essential data or reagents; JB, Conception and design, Acquisition of data, Analysis and interpretation of data, Drafting or revising the article; JER, Conception and design, Analysis and interpretation of data, Drafting or revising the article; GL, Conception and design, Acquisition of data, Analysis and interpretation of data, Drafting or revising the article, Contributed unpublished essential data or reagents

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
