## [Decision Letter]

[Editors’ note: the authors performed additional work to address the concerns raised in the first round of peer review, shown below, and submitted again for further consideration.]

Thank you for sending your manuscript entitled “Inter-Golgi Transport Mediated by COPI-containing Vesicles Carrying Small Anterograde Cargo and Golgi Resident Enzymes” for consideration at *eLife*. Your article has been reviewed by 3 experts, one of whom is a member of our Board of Reviewing Editors. The reviewers were impressed with the development of a new approach to address the ongoing controversy on the mechanism of cargo transport across the Golgi stack. Your light microscopy based analysis suggests the potential involvement of COPI or related carriers in the anterograde transport of cargo. However, the reviewers felt the data was not conclusive to merit publication at this time. The reviewers had two major concerns:

1) The data are derived from the experiments performed at 20**°**C. This is non-physiological and could create missorting because of accumulation of over-expressed cargo at the TGN. You will have to find a way out of this situation and perform the experiments at 32–37**°**C.

2) The data do not rule out the possibility that the intermediates (COPI carriers?) being visualized are en route to or from another donor or acceptor compartment.

We realize that addressing these two issues experimentally will require considerable effort. But until the identity of the visualized vesicles is determined, we feel your manuscript is currently not suitable for further consideration at *eLife*.

---

## [Author Response]

*1) The data are derived from the experiments performed at 20***°***C. This is non- physiological and could create missorting because of accumulation of over-expressed cargo at the TGN. You will have to find a way out of this situation and perform the experiments at 32*–*37***°***C*.

In order to assess if inter-Golgi transport of anterograde cargo could occur at 37**°**C, we co-expressed a dominant interfering mutant of PKD, known to block TGN exit (Baron et al., 2002) within the fused cell. The presence of the PKD mutant did not alter the inter-Golgi exchange of the small anterograde cargo that could be monitored at 32**°**C in this condition. This is now reported and discussed in the manuscript (Figure 8 and related text).

We realize that our protocol may alter the protein distribution through the Golgi stack and we cannot claim that resident enzymes are exclusively transported via a retrograde pathway, whereas the anterograde cargo is exclusively transported along a forward track. However if the transport that we observed was completely emanating from a unique mode of transport (retrograde transport from the TGN), both types of cargo should always be loaded into the same carriers. This is obviously not the case (Figure 7). Often, cargoes were segregated into different carriers, supporting a “selective bi-directional” transport. This is now discussed in the revised manuscript.

*2) The data do not rule out the possibility that the intermediates (COPI carriers?) being visualized are en route to or from another donor or acceptor compartment*.

In the first draft we already ruled out a major involvement of ER (no co- localization of cargo with COPII; inter-Golgi exchange was not affected by overexpression of Sar1 mutant, nor by H89 treatment). Now we show that the anterograde cargo does not co-localize with Rab5 and Rab7, which labeled early and late endosomes, respectively (Figure 4–figure supplement 1). This rules out that the inter-Golgi exchange is indirectly occurring via transport through the endosomal network. Together with our FRAP experiments that allow Golgi to be the only source of fluorescence, our results strongly suggest that we are monitoring a direct exchange between separated Golgi.